# Dependence of Skin-Electrode Contact Impedance on Material and Skin Hydration

**DOI:** 10.3390/s22218510

**Published:** 2022-11-04

**Authors:** Krittika Goyal, David A. Borkholder, Steven W. Day

**Affiliations:** 1Department of Microsystems Engineering, Rochester Institute of Technology, Rochester, NY 14623, USA; 2Department of Biomedical Engineering, Rochester Institute of Technology, Rochester, NY 14623, USA

**Keywords:** dry electrodes, in-home monitoring, biopotential signals, electrode material, skin-electrode contact impedance

## Abstract

Dry electrodes offer an accessible continuous acquisition of biopotential signals as part of current in-home monitoring systems but often face challenges of high-contact impedance that results in poor signal quality. The performance of dry electrodes could be affected by electrode material and skin hydration. Herein, we investigate these dependencies using a circuit skin-electrode interface model, varying material and hydration in controlled benchtop experiments on a biomimetic skin phantom simulating dry and hydrated skin. Results of the model demonstrate the contribution of the individual components in the circuit to total impedance and assist in understanding the role of electrode material in the mechanistic principle of dry electrodes. Validation was performed by conducting in vivo skin-electrode contact impedance measurements across ten normative human subjects. Further, the impact of the electrode on biopotential signal quality was evaluated by demonstrating an ability to capture clinically relevant electrocardiogram signals by using dry electrodes integrated into a toilet seat cardiovascular monitoring system. Titanium electrodes resulted in better signal quality than stainless steel electrodes. Results suggest that relative permittivity of native oxide of electrode material come into contact with the skin contributes to the interface impedance, and can lead to enhancement in the capacitive coupling of biopotential signals, especially in dry skin individuals.

## 1. Introduction

Accurate and reliable in-home monitoring of biopotential signals such as electrocardiogram (ECG) can provide early diagnosis of cardiac health conditions [1,2]. Clinically, ECG is performed using wet electrodes, which contain a conductive gel between the skin and the Ag/AgCl metal contact. The mechanistic principle is based on an electrode-electrolyte interface’s electrochemical reactions, which shows that resistive coupling is dominant due to the charge transfer phenomenon [3]. The presence of gel also moistens the skin, which helps in lowering the skin-electrode impedance [4]. Wet electrodes provide high signal quality, however are single-time use, need skin preparation, and can cause skin irritation. In addition, Huigen et al. [5] demonstrated that the main origin of noise in surface electrodes originates in the electrolyte-skin interface and highly depends on the electrode gel. Though wet electrodes exhibit excellent signal quality for short duration recordings, they are not suitable for long-term monitoring as the gel dries over time, which leads to signal degradation.

Dry electrodes have the potential to overcome all the challenges associated with wet electrodes; they are easy to use, reusable, and do not lead to discomfort. However, dry electrodes have an unstable electrochemical interface due to the absence of gel, which results in high and variable contact impedance, drift, and therefore low repeatability. Skin-electrode contact impedance plays a significant role in the performance of dry electrodes and a low skin-electrode contact impedance is desirable for the acquisition of a high-quality signal [6]. To achieve a lower skin-electrode contact impedance, it is important to understand the signal transfer mechanism occurring at the interface and the factors contributing to the skin-electrode contact impedance. In dry electrodes such as made from stainless steel, no actual charge crosses the electrode/electrolyte interface, as it is an inert metal and difficult to oxidize or dissolve [7]. Instead, a displacement current is present as a result of capacitance at the interface. Moreover, in the absence of any gel or sweat on the skin, the behavior of metal electrodes made from inert materials can be considered capacitive transducers [8]. However, it is necessary to understand how the signal is capacitively coupled from the body to the sensing electrodes. In this work, the dry electrode model is considered to act like a capacitive transducer. The heart causes immediate changes in the electric potential within the tissue, which is sensed by the metal electrodes. The metal electrode is thought to act as one plate of a capacitor, with the deeper tissue layers as the other plate of the capacitor. Thus, in our hypothesis, we considered the thin metal native oxide, air gaps, and the dry outer layer of the skin (stratum corneum) together to act as a dielectric.

For skin-based biopotential signals, the topmost layer of the skin, the stratum corneum, is important as the impedance of this layer is a major portion of the total skin impedance, and it dominates the skin impedance in the low-frequency range (1 Hz to 10 kHz) [9]. In the absence of electrolyte/sweat, the higher stratum corneum impedance makes the acquisition of high-quality signals with dry electrodes very challenging, especially in people with dry skin. Researchers have tried to overcome the high stratum corneum impedance by developing micro tips/pin-based contact electrodes [10], where the surface of the electrodes coming in contact with the skin consists of pins/spikes, with a height of the order of 150 μm [11], which is sufficient to penetrate the 20 μm-thick stratum corneum. Such an approach overcomes the high impedance offered by the stratum corneum and the charge transfer mechanism is similar to that of wet electrodes [10]. One of the major drawbacks of such an approach is it can lead to skin irritation and infection, as the pins pierce the skin and come into contact with the fluids underneath the stratum corneum [11]. Some other types of dry electrodes have been developed with polymer coatings to enhance the electrochemical performance [4,12]. When left for a certain amount of time on the skin, they are likely to perform similarly to wet electrodes for two reasons. First, the impedance of the stratum corneum is greatly affected by changes in hydration. Due to the accumulation of perspiration under the electrodes, it hydrates the skin and the capacitance of the interface improves. Second, the sweat acts as the electrolyte and facilitates the transfer mechanism similar to that of wet electrodes. One major drawback of such types of electrodes is that fabrication is complex and polymers may flake off [4].

To overcome the challenges of higher skin-contact impedance, researchers have also tried increasing the area of the electrode [12,13]. However, increasing the area alone to achieve lower skin-electrode contact impedance has a limited application, and is not applicable for miniature devices such as watches. Other efforts include using different electrode materials to improve the performance of dry electrodes [14,15]. The classical approach followed by researchers for characterizing different electrode materials is entirely empirical and consists of carrying out experiments on human skin with different chosen materials followed by qualitative or quantitative comparison with the wet electrodes (gold standard) [14,15,16]. Subsequently, the most satisfactory material based upon the performance is chosen. One of the major limitations of this empirical approach is the inter- and intra-subject variability in the skin properties. Additionally, some researchers have used the equivalent electrode-skin impedance modeling and compared the wet and dry electrode performance [17], and demonstrated the decrease in impedance corresponding to the increase in the electrode area [12]. However, the role of electrode material (typically metal-based) in the skin-electrode interface has not been studied and the rationale behind metal-based electrode material performance in the dry case scenario is not yet understood [14,15]. Thus, there is a need to understand the optimal set of solutions that can help in designing in-home monitoring devices effectively. This understanding will help to optimally design the electrodes for applications such as a watch with a miniature area, people with dry skin, and measurement locations where the signal amplitude is weak (<0.1 mV for buttocks ECG) such as measurement from the back of the thigh using a toilet seat [18].

The present work focuses on investigating the electrode-skin impedance dependence on the electrode material, area, and skin hydration by developing a skin-electrode interface circuit model. The circuit model includes aspects of prior models using a network of passive electrical circuit elements and incorporates a rationale for the electrode material. Parameter values of components within the model were determined by fitting empirical data from controlled benchtop experiments on a skin phantom designed in our previous work [19]. The phantom simulates the effect of skin hydration in a controlled predictive manner and mimics the electrical properties of the skin in the low-frequency range. The circuit parameters were fitted and computed to understand the effect of variables such as material, area, and hydration status in the skin-electrode contact impedance. Validation was performed using in vivo human subject impedance data captured across ten normative subjects using impedance spectroscopy. Lastly, ECG was acquired using different dry electrode configurations, integrated into a toilet seat cardiovascular monitoring system, and the ability to capture high-ECG signal quality is demonstrated.

## 2. Materials and Methods

### 2.1. Electrode Material

Several metal-based candidate materials comprising stainless steel, titanium, copper, silver, nickel, gold, and brass were explored for consideration as electrode materials. The selection criteria included: high electrical conductivity, high corrosion resistance, hypoallergenic, robust to abrasion, and chemical resistant. Of the materials evaluated, stainless steel 304 (SS) and titanium grade 2 (Ti) were selected as the most readily available materials.

The electrical conductivity of the selected materials was investigated using a four-point probe setup (Model RM 3000, Jandel Engineering Ltd., Leighton Buzzard, UK). The surface roughness of the materials was measured using a non-contact optical profilometer (ST400, Nanovea Inc., Irvine, CA, USA) to investigate the effective contact area coming in contact with the skin and quantitative measures of surface roughness such as the arithmetic average of the roughness profile (Ra) were obtained. SS is an alloy that contains iron, chromium, and nickel [20], thus the composition of native oxides was performed using auger electron spectroscopy (AES) to examine the composition of the oxides present on the stainless surface that directly contacts the skin. In addition, ion sputter etching was performed to evaluate the composition as a function of depth. AES was also performed across the titanium sample to investigate the composition of the native titanium oxide [21].

### 2.2. Skin-Electrode Impedance Equivalent Model for Dry Electrodes

#### 2.2.1. Impedance Measurements on Phantom

Skin-electrode contact impedance was investigated by carrying out impedance measurements on the phantom developed in previous work [19]. To summarize, the fabricated phantom comprises two layers representing the deeper tissues and stratum corneum. The lower layer simulated deeper tissues and the hydration of the stratum corneum was modeled in a controlled way by varying porosity of the phantom’s upper layer. Impedance spectroscopy measurements were performed with a potentiostat (Reference 600, Gamry Instruments Inc., Warminster, PA, USA) in a two-electrode configuration, where the working (W) and counter (C) probe carried the current, and working sense (WS) and reference (R) measured voltage. Electrodes were fabricated from a 0.3 mm-thick sheet, as this was easy to cut. Different electrode configurations consisted of electrode material: stainless steel and titanium with electrode area: 4 cm^2^ and 9 cm^2^, each arrangement tested in a controlled manner on a set of phantoms simulating dry and hydrated skin. The hydration state modeled with 0.16% porosity simulated dry skin and that with 0.28% mimicked hydrated skin impedance. Using the potentiostat, a frequency sweep of 0.1 Hz–100 kHz was performed and an impedance response was obtained.

#### 2.2.2. Development of Model

A simplistic model for metal-based dry electrodes consisting of a circuit model of the electrode and the skin (Figure 1) was developed using Gamry Echem Analyst software. Deeper tissues were represented by a resistor (*R_d_*), and the stratum corneum by a capacitor (*C_sc_*) and resistor (*R_sc_*) in parallel. The contact interface between the skin and electrode, including the thin native oxide, air gaps due to the skin roughness, stiff metal, and the stratum corneum interface was considered to act as a dielectric in the transduction mechanism and modeled by a contact capacitor (*C_c_*). Thus, the electrode was thought to act as one of the capacitor plates, and the interface located at the stratum corneum was considered as the other capacitor plate. Both the relative permittivity (also known as dielectric constant) and the thickness of the native oxide layer are material-dependent. The air gap depends on the pressure and effective contact between the electrode and the skin. The dielectric properties of the stratum corneum depend on skin type, part of the body, hydration level, and are highly subject-dependent. Thus *C_c_* was considered to vary as per Equation (1), where the capacitance is directly proportional to the relative permittivity and area, inversely proportional to the distance between two plates:(1)$$C=\frac{\varepsilon_{0}\varepsilon_{r\;}A}{d}$$
where *ε*_0_ is the relative permittivity of free space, *ε_r_* is the relative permittivity of a material, *A* is the area of the plate, and *d* is the distance between the two plates.

The parameters representing the stratum corneum, *C_sc_* and *R_sc_* were modeled for dry and hydrated phantom similar to that of dry and hydrated skin. To obtain good contact between the electrode and the phantom’s surface, a micromanipulator (M325, World Precision Instruments, Sarasota, FL, USA) was used to hold the electrode against the phantom’s surface, as the effective contact between the electrode and the skin affects the interface impedance significantly [22]. The micrometer head was adjusted until it just touched the snap-on connector in order to hold the electrode against the phantom’s surface. Thus, this ensured a consistent and very low pressure between the electrode and phantom’s surface. The capacitive impedance holds an inverse relation to capacitance given by Equation (2) and substituting the value of C, using Equation (1), results in Equation (3):(2)$$Z_{c}=\frac{1}{\omega C}$$
(3)$$Z_{c}=\frac{d}{\omega \varepsilon_{0}\varepsilon_{r\;}A}\;$$
where *ω* = 2*πf*, *ω* is in rad/s, and *f* is the frequency (Hz).

The model (Figure 1) was used to fit the impedance response obtained across various electrode configurations comprising different electrode materials and sizes, tested on a dry and hydrated phantom. For understanding the physics and the mechanism, the stratum corneum and the contact interface components were thought to be capacitors, as they have a physical meaning in the real world. However, a constant phase element was used for the fitting, as it accounts for factors such as non-uniform current distribution and surface roughness [17,20]. The constant phase element is represented by Equation (4), where *CPE* represents the capacitance, *ω* = 2*πf*, ω is in rad/s, f is in Hz, *α* is the exponent, and lies between (0 < *α* ≤ 1). From Equations (2) and (4), When *α* = 1, the constant phase element becomes equivalent to a capacitor. The modeling parameters representing deeper tissue layers (*R_d_*), stratum corneum (*R_sc_*, *CPE_sc_*) and contact interface (*CPE_c_*) were computed by fitting the obtained impedance response. The fitting of the circuit parameters was restricted to lower and upper bounds, to avoid physically unreasonable values [6]. As per values found in the literature, *R_d_* was assigned 100 Ω–10 kΩ, *R_sc_* varied in the range of 10 kΩ–10 MΩ, and *CPE_sc_* ranged from 1–500 nSs^α^ (S and s represent Siemen and seconds, respectively) with 0 < *α* ≤ 1 [6,17]. *CPE_c_* was assigned a wide range of 1–20,000 nSs^β^ with 0.5 < *β* ≤ 1. The *C_sc_* was calculated by substituting the obtained values for *R_sc_*, *CPE_sc_,* and α in Equation (5). *C_c_* was calculated by using Equation (6), where *CPE_c_* along with exponent β represents the constant phase element of the interface and $\omega_{max}$ is the frequency at which the imaginary part of impedance reaches maximum [23]. The error associated with calculated fitted parameters was investigated. The goodness of fit (chi-squared) computed after fitting the parameters provided the fitting accuracy, which can be defined as the square of the percentage error between the experimental and fit values [24]. For this work, a value of an order of (10^−2^) was considered a sufficiently good fit, which indicates a 10% difference between the experimental and fitted data [25] and a model was considered valid if the error associated with every calculated fitted parameter was smaller than its respective components.
(4)$$Z_{\mathrm{CPE}}=\frac{1}{{\left(j\omega \right)}^{\alpha }CPE}$$
(5)$$C_{\mathrm{sc}}=\frac{{\left(CPE_{sc}R_{sc}\right)}^{1/\alpha }}{R_{sc}}\;$$
(6)$$C_{c}=CPE_{c}{\left(\omega \right)}_{max}{}^{\beta -1}\;$$

### 2.3. Normative Subject Testing

#### 2.3.1. In Vivo Skin-Impedance Measurements

To evaluate the effect of SS and Ti in vivo on the skin-electrode interface, measurements were conducted across the thigh, using a toilet seat platform. It consists of a pair of SS electrodes integrated on the right side and a pair of Ti electrodes on the left side of the toilet seat (Figure 2). The human subject study was performed across ten normative subjects under informed consent and protocol approved by the Institutional Review Board at Rochester Institute of Technology. A two-electrode configuration was used, as to measure the skin-electrode contact impedance, for studying the interaction of metal dry electrodes with the skin. Potentiostat (Reference 600, Gamry Instruments Inc., Warminster, PA, USA) was used to generate a voltage signal of 25 mV RMS AC in magnitude, and a frequency sweep of 0.1 Hz–100 kHz was performed. In preliminary experiments, we confirmed the linearity of the phantom over a range of 5–25 mV, shown in the Appendix A). To ensure electrical safety, both the potentiostat and the laptop were plugged into a medical-grade isolation transformer (ILC-1400MED4, TSI Power Corp., Antigo, WI, USA). In the first recording, the leads of the potentiostat were connected to the right side of electrodes consisting of SS electrodes while the Ti electrodes were covered by a vinyl sheet. The subject was asked to sit on the seat, such that the skin of just one thigh was in contact with one set of electrodes to measure the impedance. The subject was instructed to stand up, the vinyl sheet was removed, the leads of the potentiostat were connected to Ti electrodes and the impedance measurements were acquired. It took approximately five minutes for each measurement. Impedance response corresponding to both the SS and Ti electrodes was analyzed. Each subject’s paired t-test was used to compute if the differences between the impedance responses obtained for SS and Ti were statistically significant (*p* ≤ 0.05).

#### 2.3.2. Electrocardiogram Acquisition

Immediately after recording the skin-electrode contact impedance measurements, ECG was captured using two sizes of SS and Ti electrodes fitted within a previously designed fully integrated toilet seat [18]. All the instrumentation/electronics were integrated into the toilet seat, and battery powered [18]. The ECG signal was differentially measured using two electrodes and an instrumentation amplifier with a high common-mode rejection ratio. The captured data was sent to a laptop via an onboard Bluetooth Low Energy radio. Different electrode configurations were integrated into the toilet seat and tested across all the subjects (N = 10). Electrode configurations comprised areas of 6 cm^2^, and 90 cm^2^, for both SS and Ti (Figure 3). For comparing signals to a high-quality signal, an alcohol swab was used in place of traditional wet electrodes. Two seat setups were used, one integrated with 90 cm^2^ SS and the other with 90 cm^2^ Ti electrodes. To achieve a smaller area of electrodes, 90 cm^2^ electrodes were covered with a vinyl mask, only exposing the 6 cm^2^ area (Figure 3). Preliminary experiments showed that electrodes completely covered with the vinyl sheet resulted in no ECG at all, thus indicating that there was no capacitive coupling of ECG signal occurring through the vinyl sheet. The ECG recordings start immediately when the subject sits on the toilet seat. Due to the measurement location, back-to-back ECG measurements across different electrode configurations were performed. For all the subjects, the measurements were performed in the following order: 90 cm^2^ Ti; 90 cm^2^ SS; 6 cm^2^ Ti; 6 cm^2^ SS. Lastly, alcohol swabs (AS) were placed on both the smaller area electrodes Ti 6 cm^2^ followed by SS 6 cm^2^. 90-s recordings were made for all the electrode configurations. The total duration for conducting the entire protocol for ECG recordings across each subject was around 20 min. All the measurements were conducted in the controlled lab environment and the subjects were asked to sit as still as possible.

#### 2.3.3. Signal Processing and Analysis

For the preprocessing of the raw ECG signal, five seconds of data were removed from the beginning and the ending of the 90-s acquisition. Signals were band-pass filtered using a second-order Butterworth filter (1 Hz–40 Hz) and a notch filter at 60 Hz [26]. The ECG signal’s main power lies in 5 Hz–15 Hz [27,28] thus the signal quality evaluation was performed by calculating the power spectral density of the signal in the 5 Hz–15 Hz to that of the overall signal in 5 Hz–40 Hz [28,29]. The power spectral density was calculated by Welch’s method [30] and the power was computed by approximating the area under the power spectral density curves. Thus, the power spectral density ratio (PSDR) can be defined as the proportion of the QRS power to that of overall signal power [28] as shown in Equation (7). Extremely high PSDR indicates the presence of large noisy spikes [28], thus a threshold was empirically chosen, and the signal was considered to be of analyzable quality if the PSDR was less than 0.80 (Figure 4).
(7)$$\mathrm{PSDR}=\frac{\int_{5\;\mathrm{Hz}}^{15\;\mathrm{Hz}}\mathrm{PSD}}{\int_{5\;\mathrm{Hz}}^{40\;\mathrm{Hz}}\mathrm{PSD}}$$

Signal-to-noise ratio (SNR) of the analyzable signals was computed to quantitatively compare the signal quality across all the electrode configurations. The filtered ECG signal was used and the r-peaks were identified based upon the Pan Tompkins algorithm along with ECG signal delineation using Neurokit2, a python package for signal processing [31]. After locating the r-peaks, the S wave was identified as the nearest point to the right of the R wave, where the downward deflection has a minimum. SNR was calculated by using equation (8), similar to [32]. The peak-peak signal (*V_signal_*) value was obtained by calculating the difference between the R wave (maximum) and S wave (minimum), and averaging across each ECG cycle. Based on the ECG morphology, peak-peak noise (*V_noise_*) was calculated as the difference between the maximum noise and the minimum noise in the segment after the T wave ends and before the P wave of the next ECG signal begins. (Figure 5). Thus, the noise was considered as the peak-peak variations in the segment between the two ECG cycles, excluding the P wave, QRS component, and T wave [32]. The mean was computed for the peak-peak noise across ECG cycles present in a 90-s recording. Further, the ratio of peak-peak signal and peak-peak was converted to decibels (dB). For the ECG signals that were identified as unanalyzable as their PSDR was greater than 0.80, they were assigned an SNR value of zero.
(8)$$\mathrm{SNR}=20\;\times \;\log_{10}\;\frac{V_{signal}}{V_{noise}}$$

Kurtosis has been widely used to determine the ECG signal quality [16,28,29]. Kurtosis was used as a metric to further evaluate the quality of the ECG signal across different configurations. Kurtosis is a statistical measure, as defined in Equation (9), and provides a measure of the shape of the curve. Higher kurtosis indicates the dominance of the QRS complex and lower indicates the absence of the QRS complex. Before calculating the kurtosis, median filtering was performed on the processed ECG signal, to remove the large spikes, as kurtosis is sensitive to outliers [33]. Kurtosis was calculated across a window of 10 seconds and an average was computed.
(9)$$\mathrm{Kurtosis}=\frac{1}{n}\sum_{i=1}^{n}[\frac{x_{i\;}-{µ}_{\;}}{\sigma }{]}^{4}$$
where µ is the empirical estimate mean of *x_i_*, and σ is the empirical estimate standard deviation of *x_i_*.

## 3. Results

### 3.1. Electrode Material Characterization

The electrical conductivity obtained using four-point probe measurements for Ti and SS resulted in 2.17 × 10^6^ S·m^−1^ and 1.30 × 10^6^ S·m^−1^, respectively. This was in accordance with the literature where the conductivity of Ti is reported as 2.38 × 10^6^ S·m^−1^ and SS as 1.45 × 10^6^ S·m^−1^ [34]. Thus, both the SS and Ti were found to be highly electrically conductive. Surface roughness measured using an optical profilometer resulted in an arithmetic average of roughness profile (R_a_) to be 1.3 microns for the Ti and 1.6 microns for the SS. This indicates that the surface of both the materials comprising native oxide are similar in smoothness, and it can be considered that the effective area in contact with the body is approximately the same for both materials. The auger spectrum for SS as a function of kinetic energy shows that only iron oxide was present in the SS native oxide along with carbon and oxygen (Figure 6). Thus, the peak at 711 eV indicated the presence of iron oxide (Fe_2_O_3_) [35]. After depth profiling, it was found that the concentration of carbon and oxygen decreased, chromium and nickel became visible, and iron increased. For titanium, the AES spectrum revealed the presence of titanium along with carbon and oxygen and the concentration of titanium increased after depth profiling. The peak at 420 eV indicated that the oxide composition comprises TiO_2_ [36].

### 3.2. Skin-Electrode Contact Impedance Model Parameters

The Bode plot showing the impedance response with respect to frequency resulted in a lower impedance for the 9 cm^2^ electrode as compared to the 4 cm^2^ electrode (Figure 7a). This indicates that the higher-area electrode leads to a lower impedance and matches with the previously reported work [12,13,37]. The equivalent impedance model was found to have a good fit to the experimental data, shown as a solid line in Figure 7a) for both 4 cm^2^ and 9 cm^2^ area electrodes tested on a dry skin phantom. The modulus of impedance for both of them was observed to have a similar slope and the impedance of the 4 cm^2^ electrode was 2 times higher as compared to that of a 9 cm^2^ electrode. This is in complete agreement, as the area of the electrode (9 cm^2^) is approximately twice the area of the other electrode (4 cm^2^). Thus, the developed phantom was found to scale the effect of impedance to that of the electrode area. The phase response for both 4 cm^2^ and 9 cm^2^ depicted a resistive behavior and higher frequencies and a capacitive behavior at lower frequencies (Figure 7b). The fit model was within 5 degrees for both 4 cm^2^ and 9 cm^2^ electrodes in the low-frequency range of 1 Hz–1000 Hz. Further, the computed parameters obtained after fitting the model (Table 1) indicated that the *R_sc_* reduced by two times and both *C_sc_* and *C_c_* increased by 2.5 times. This is consistent with the mathematical relationship, as resistance is inversely proportional to area and capacitance is directly proportional to the area, thus clearly depicting the effect of area on the skin-electrode contact impedance (Table 1). *R_d_* was close for both 9 cm^2^ and 4 cm^2^ electrodes, indicating the phantom properties were stable during the course of experiments.

The bode plots in Figure 7c show the normalized impedance with respect to area and depict the impedance response corresponding to different electrode materials, on a dry and hydrated phantom. The impedance response for the Ti electrode was lower than that of the SS electrode (Figure 7c). The area-normalized electrode-skin impedance at 10 Hz for SS and Ti was 2.40 and 1.70 MΩ cm^2^ on a dry phantom. The fitted phase response of Ti and SS on both dry and hydrated phantom was within 10 degrees of that of experimental data. The model fitting parameters *R_d_*, *R_sc_*_,_ and *CPE_sc_* were fixed during the fitting of the impedance curve for SS and Ti for a dry phantom, and *CPE_c_* indicated the effect of electrode material on the skin-electrode interface. *C_c_* for Ti (267 nF cm^−2^) was 2.3 times higher than that of SS (114 nF cm^−2^), indicating better capacitive coupling for Ti than SS (Table 2).

With the change in the hydration status of the phantom, an impedance of an order of magnitude lower was found for the hydrated phantom as compared to the dry phantom (Table 2; Hydrated phantom, *R_sc_* (3.0 MΩ cm^2^) was lower, and *C_sc_* (190 nF cm^−2^) was 4.6 times higher than the dry phantom (6.4 MΩ cm^2^; 41 nF cm^−2^)). The combination of *R_sc_* and *C_sc_* lowered the total impedance by an order of magnitude, which indicates that the stratum corneum properties have a significant contribution to the total skin impedance. *R_d_* was fixed for the dry and hydrated phantom, as the novel porosity-based approach allows changes to the properties of the stratum corneum in a controlled way. The contact capacitance (*C_c_*) for titanium on a hydrated phantom (420 nF cm^−2^) was approximately 2 times higher than that on a dry phantom (267 nF cm^−2^). This indicates that the hydrated skin phantom not only provided a lower stratum corneum impedance but also improved the contact impedance by two times as compared to the dry phantom. Further, it is worthwhile to mention that all the impedance graphs, irrespective of the dry electrode configurations being tested, have a consistent slope, indicating a similar transduction mechanism at the interface.

The fitted values of *CPE_sc_*, *CPE_c_, α* and β are provided in Appendix A. The value of *α* was 0.9, similar to the prior research works [38,39], and indicated roughness of skin and non-linear distribution of time constants. The value of β in the range of 0.5–0.6 can be attributed to dissipative effect at the electrode-skin interface [17]. For ease of understanding, the values were converted and presented in terms of real capacitance in Table 1 and Table 2. The error associated with every calculated fitted parameters was substantially smaller than the calculated value of components (Appendix A). Across the range of frequency, the model agrees with the acquired data within 10% of the value. Both Table 1 and Table 2 indicate the chi-squared value ranging from 0.7 × 10^−2^ and 4.4 × 10^−2^ for the fitting of the parameters to the circuit model. The obtained goodness of fit values is of the same order of magnitude as that of previous research works that depicted a good agreement between the fitting and experimental data [6,40].

### 3.3. Normative Subject Testing

#### 3.3.1. In Vivo Skin-Electrode Contact Impedance

In vivo impedance measurements captured across the thigh resulted in higher skin-electrode contact impedance for SS as compared to Ti (Figure 8). The impedance spectrum was obtained from 0.1 Hz–100 kHz, but for comparison normalized impedance at 10 Hz is shown in Figure 8. FDA guidelines suggest 10 Hz as the reference frequency to validate the performance of recording electrodes [40,41]. The mean area-normalized electrode-skin impedance at 10 Hz for SS and Ti was 2.68 and 2.06 MΩ cm^2^, respectively. SS resulted in a high variance in skin-electrode contact impedance, compared to Ti, although a paired *t*-test (*p* = 0.032) indicated that the contact impedance for Ti is significantly lower than that of SS. Further, each of the five individuals above the median (blue line in Figure 8), were observed to have significantly lower impedance for Ti as compared to SS, whereas the impedance for the other five individuals below the median was found not to be significantly different.

#### 3.3.2. ECG Demonstration

Figure 9 shows an example of ECG segments captured across one subject with different electrode configurations and their corresponding SNR. Both Ti 6 cm^2^ alcohol swabs (AS), shown in black, and SS 6 cm^2^ alcohol swabs (AS), shown in red, depicted high SNR of 15 dB and 13 dB, respectively. Ti 90 cm^2^, shown in blue, resulted in higher SNR as compared to SS-90 cm^2^, shown in green. Both Ti-6 cm^2^, shown in yellow, and SS-6 cm^2^, shown in cyan, exhibited lower SNR in comparison to larger area electrodes. Noise was significantly visible in the smaller area electrode configurations.

SNR computed for different electrode configurations across 10 subjects was significantly different for SS 90 cm^2^ and Ti 90 cm^2^ electrodes with a value of 0.011 (*p* < 0.05, Figure 10a). Higher SNR obtained for Ti 90 cm^2^ to that of SS 90 cm^2^ indicates better performance of Ti for the ECG acquisition. Data points corresponding to each subject are represented by a different shape and color and the box plot shows the entire distribution with a blue line indicating the median and whiskers describing entire the range of values. For alcohol swabs placed on SS and Ti, SNR was observed not to be significantly different, indicating the contribution of alcohol swabs towards capacitive coupling more than that of electrode material. For both SS and Ti 6 cm^2^ electrodes, analyzable ECG could not be obtained for 7 subjects, hence SNR was assigned zero as depicted in Figure 10a. This indicates the smaller area electrodes were not sufficient for capturing ECG signals across seven out of ten subjects. For the three subjects, a similar trend from a material perspective was observed, where Ti resulted in higher SNR as compared to SS electrodes.

Kurtosis resulted in higher values for Ti 90 cm^2^ as compared to SS 90 cm^2^, thus indicating the dominance of QRS peaks in Ti as compared to SS (Figure 10b). Kurtosis for alcohol swabs on SS and Ti was not significantly different. Kurtosis for Ti 6 cm^2^ and SS 6 cm^2^ was lower than that of other electrode configurations. Moreover, the data points corresponding to the subjects whose PSDR was higher than 0.8 and assigned an SNR of zero, resulted in extremely low and negative kurtosis, which indicated the lack of QRS complex. Further, the evaluation of the contribution of signal and noise components to SNR revealed that the noise obtained with SS (90 cm^2^) and Ti (90 cm^2^) electrodes was significantly different, however, the signal was not significantly different (Figure 11). This indicates that the contribution of the area was large enough to lower the skin-electrode impedance, and capture the ECG signal for both SS and Ti. However, low-frequency noise integrated into the 1 Hz–40 Hz frequency band was higher for SS as compared to Ti. This is because contact impedance was higher for SS, thus increasing the noise level. This is in agreement with the findings of Kappenman et al., that high-electrode impedance may not contribute to the attenuation of the signal, but may increase the noise level [42].

## 4. Discussion

In this work, a simple mechanistic circuit model has been proposed to isolate and understand the role of the electrode area, material of a dry electrode on dry and hydrated skin, while isolating the influence of each variable independently. Similar to prior work [43], an analogous RC circuit model was used to simulate the electrical model of the skin. Our method of using empirical data to determine parameter values showed the effect of area, which resulted in a lower skin-electrode contact impedance with an increase in the area, which corresponds to prior work [12,37] and theoretical predictions, giving us confidence in the method. Thus, this method was used to investigate the interface properties using the controlled skin properties via a phantom. Table 2 provides the individual contribution of contact impedance and stratum corneum in the capacitive coupling mechanism. Because contact impedance comprises native oxide, the electrical and dielectric properties of the native oxides were investigated. The Auger peaks (Figure 6a) obtained for SS at 711 eV showed that (Fe_2_O_3_) dominated the composition of SS as the native oxide. The peak at 420 eV (Figure 6b) revealed that the oxide composition for Ti was comprised mainly of titanium oxide (TiO_2_). Therefore, the thickness, conductivity, and relative permittivity of the native oxide layers were investigated. The native oxide thickness of titanium and stainless steel was found to be a few nanometers thick, 3 nm for that of Ti [44] and 5 nm for SS [45]. Boxley et al. showed the conductivity of TiO_2_ to vary between 10^−3^ to 10 Sm^−1^ [44]. This is similar in order of magnitude to that of conductivity of Fe_2_O_3_ to ~1.3 × 10^−3^ Sm^−1^ [46]. The relative permittivity of titanium native oxide was found to be 4 times higher for Ti native oxide (TiO_2_, *ε_r_* = 60) [47], than that of SS native oxide (*ε_r_* = 15.6) [48]. The fitted curve and the computed contact capacitance (*C_c_*), reveal that the Ti shows enhanced capacitive behavior (×2.3 times) as compared to SS (Table 2), therefore lower contact impedance (Equation (2)). Thus, this can be attributed to the higher relative permittivity of Ti native oxide to that of SS native oxide coming in contact with the skin. Moreover, there have been attempts made by researchers to use aluminum as a metal-based dry electrode for recording biopotential signals, and poor performance has been reported for aluminum [49,50]. On looking into the relative permittivity of the native oxide for aluminum (*ε_r_* ~9) [48], it suggests a poor performance compared to that of SS and Ti. This further validates the mechanism of capacitive coupling for metal-based dry electrodes.

Stratum corneum properties play a significant role in the capacitive coupling transfer mechanism [38,51]. The fitted parameters for stratum corneum (*R_sc_*, *C_sc_*) on a dry phantom (Table 2), were found to be of the same order of magnitude similar to that of literature, where resistance per unit area is of the order of 10^5^ Ω cm^2^ (ranging between 0.1–5000 kΩ cm^2^) and capacitance per unit area is of the order of 30 nF cm^−2^ [52]. Interestingly, we observed that for the hydrated phantom, *C_sc_* increased by 4.5 times, as compared to a dry phantom (Table 2). This can be attributed to the fact that the dielectric properties of the stratum corneum are significantly affected by hydration. S. Björklund et al. showed that the relative permittivity of dry stratum corneum is 10 and that of hydrated SC is 49 [39]. Table 2 indicates that *C_c_* was approximately 2 times more for the hydrated phantom than for a dry phantom. This can be attributed to the improved contact, as in a hydrated state due to higher porosity, saline solution is filling the air gap and acting as the junction liquid (*ε_r_* = 80), thus the effective contact capacitance is improved between the electrode and stratum corneum. Thus, for the hydrated state, the changes in skin-electrode contact impedance due to hydration were significant as compared to the electrode material.

Since hydration is the most significant factor in contributing toward a lower skin-electrode impedance, nowadays, progress is made in the field of semi-dry electrodes [53]. The presence of a low amount of electrolyte as a liquid rather than conductive gel overcomes the limitations of the wet electrodes such as the need for skin preparation/abrasion of the skin and drying out of the gel [54]. The discharge of a small amount of electrolyte can achieve a local skin hydration effect, which results in a lower skin-electrode impedance [55]. Such type of semi-dry technology has great potential especially in recording high-density electroencephalography (EEG) signals. For recording EEG signals with the typically used semi-dry electrodes, electrode material might not play a substantial role as hydration is the most significant factor in achieving a low skin-electrode impedance. However, for recording ECG signals, in some applications such as smart watches (Apple watch), hand-held monitors (AliveCor Kardia Mobile), and toilet seat-based cardiovascular monitoring system (Casana), the electrodes used are completely dry because adding the electrolyte prior to each measurement is not practical. Thus, investigation in this work was carried out to study the role of electrode material in lowering the electrode-skin impedance in the absence of added electrolyte.

Our results depicted higher SNR with Ti as compared to SS (Figure 10A), which is in agreement with the findings of Meziane et al. [14]. However, Meziane et al. demonstrated the better performance of Ti as compared to SS only through the empirical biopotential signal acquisition across human subjects. In our work, we further investigated the skin-electrode contact impedance both on a phantom and human skin along with ECG acquisition. For both the phantom and human skin, contact impedance was found to be lower for titanium than SS (Figure 7c and Figure 8). This is in agreement with Searle et al. where lower contact impedance against time was observed for Ti as compared to SS, however was not significant when compared across 5 subjects [56]. In our work, we noticed that the skin-electrode contact impedance was not very different for subjects below the median line (Figure 8). Based on this, the data was divided into two subpopulations (dry and hydrated) and the correlation with SNR corresponding to the ECG signal was investigated. Both skin-electrode contact impedance (Figure 12a) and SNR (Figure 12b) obtained with different materials were significantly different for dry skin and not for hydrated skin individuals. This indicates that the material plays a significant role for the dry skin individuals, rather than people with hydrated skin. Additionally, dry skin people with higher skin impedance are more sensitive to noise pickup. Thus, the higher relative permittivity of TiO_2_ indicates a lower capacitive impedance in comparison to SS native oxide. These results suggest that the relative permittivity at the interface of skin and the metal plays a significant role in the enhancement of the capacitive coupling phenomenon. Moreover, it was evident that for individuals characterized as having hydrated skin, an analyzable ECG signal was even detected with the small area (6 cm^2^) electrodes (Figure 10a), thus further indicating the presence of a lower skin-electrode contact impedance.

One of the limitations of this study includes that unlike simultaneous ECG measurements performed in earlier work [14] to compare the wet and dry electrode performance, we compared the back-to-back ECG measurements, due to the measurement location. Secondly, an alcohol swab was placed on the electrode for capturing a high quality to compare signals and considered a gold standard. Excellent signal quality was observed with alcohol swabs, however, the mechanistic principle involved in the working of the alcohol swabs is very different from wet electrodes. We believe the mechanistic principle for the functioning of the alcohol swab is based upon improved capacitive coupling due to the increase in the effective area and the presence of higher relative permittivity between the skin and the metal electrode. Due to the skin’s rough surface and the stiffness of the metal, the electrodes are not in complete contact with the skin; hence, by placing the alcohol swab between the skin and electrode, the air gaps are replaced by fluid, which leads to an increase in capacitive coupling ($\varepsilon_{r}$ for air is 1 and that of alcohol swab made from 70% IPA is 35) [57]. Moreover, along with increased effective contact area and higher relative permittivity, it also moistens the skin, thus lowering the stratum corneum impedance. The proposed rationale is in complete agreement with previously reported results where the dry electrode was wetted with polyethylene glycol ($\varepsilon_{r}$ ~10) and water ($\varepsilon_{r}$ ~80) [58].

The findings of this work suggest that investigating dielectric properties, such as thickness, and relative permittivity of the native oxide, can be one of the approaches when selecting material for biopotential electrodes. Higher relative permittivity of the electrode material native oxide can lead to an enhancement in the capacitive coupling phenomenon and thus a lower skin-electrode impedance. An increase in electrode area can reduce skin-electrode impedance (Figure 7a), but for applications where the area is a limiting factor, such as watches, electrode material selection might help in achieving a lower electrode-skin impedance (Figure 7c). For applications such as a toilet seat cardiovascular monitoring system, an optimal combination of both area and material can help in achieving lower skin-electrode impedance and thus higher signal quality (Figure 10). Titanium has a potential to be used as an electrode material for achieving higher signal quality, which can be accounted to the higher relative permittivity of Ti native oxide. This work indicates that the approach of prioritizing the relative permittivity of the oxide layer in material selection has the potential to be used as a guide to material selection. Moreover, for dry skin individuals, Ti depicted a lower variance than SS (Figure 8), which indicates higher repeatability with Ti than with SS. This can be leveraged, as the properties of humans are variable, which leads to skin impedance changes over time. Thus, titanium-based metal electrodes as compared to SS can improve the reliability of in-home monitoring systems.

## 5. Conclusions

This work explores the role of electrode material and skin hydration in a controlled way on a phantom using a skin-electrode interface model. The model demonstrates the significance of electrode material in the mechanistic principle involved in the working of the dry electrode for biopotential signal acquisition. The developed model studies the contribution of each variable individually and can be used to provide insight into the development of future electrodes. The results of the model suggest the relative permittivity of the native oxide for the electrode material can provide an insight into the electrode performance for biopotential measurements. The ECG acquisition conducted across human subjects indicates titanium as a superior material for the acquisition of biopotential signals with less noise as compared to stainless steel, especially for dry skin individuals. Thus, titanium compared to stainless steel has the potential to improve dry electrode performance. This work provides insight into the transduction mechanism of the metal-based dry electrodes, and the significance of native oxide and skin dielectric properties in the electrode performance, which can help in the evaluation of other electrode materials.

## Figures and Tables

**Figure 1 sensors-22-08510-f001:**
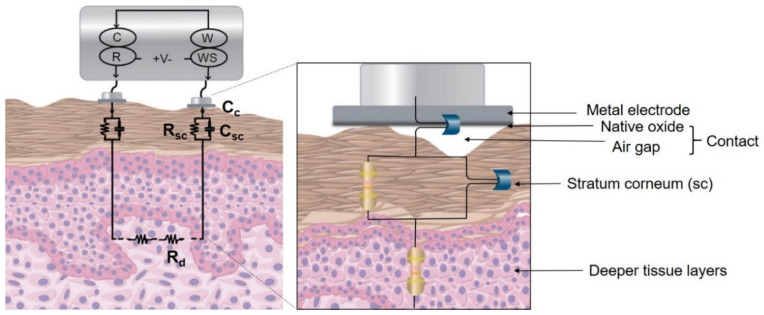
Two-electrode setup and an equivalent skin-electrode model for metal-based dry electrodes. Two- electrode configuration setup is shown where C, R, W, and WS represent counter, reference, working, and working sense leads of the potentiostat, respectively. An analogous electrical equivalent circuit is shown where contact is represented by a capacitor (*C_c_*), along with the stratum corneum as a parallel combination of resistor (*R_sc_*) and capacitor (*C_sc_*), and deeper tissue layers as a resistor (*R_d_*). The zoomed-in view represents the air gaps due to the surface roughness of the stratum corneum, the thin native oxide that comes in direct contact with the skin.

**Figure 2 sensors-22-08510-f002:**
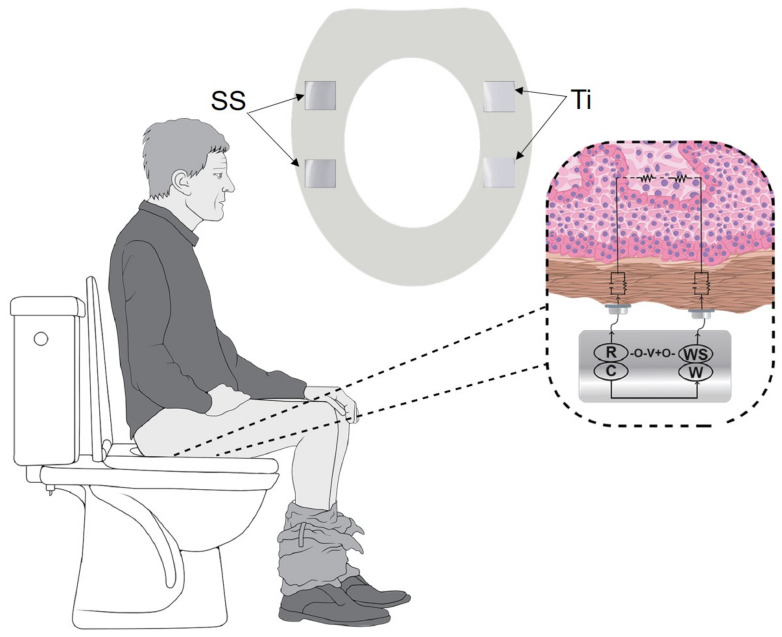
Experimental setup for acquiring the skin-electrode contact impedance measurements. Stainless steel (right side) and titanium electrodes (left side), 9 cm^2^ each, were integrated into a toilet platform. The zoomed-in view of the skin cross-section coming in contact with the two electrodes along with the electrical equivalent model of the electrode-skin interface is shown.

**Figure 3 sensors-22-08510-f003:**
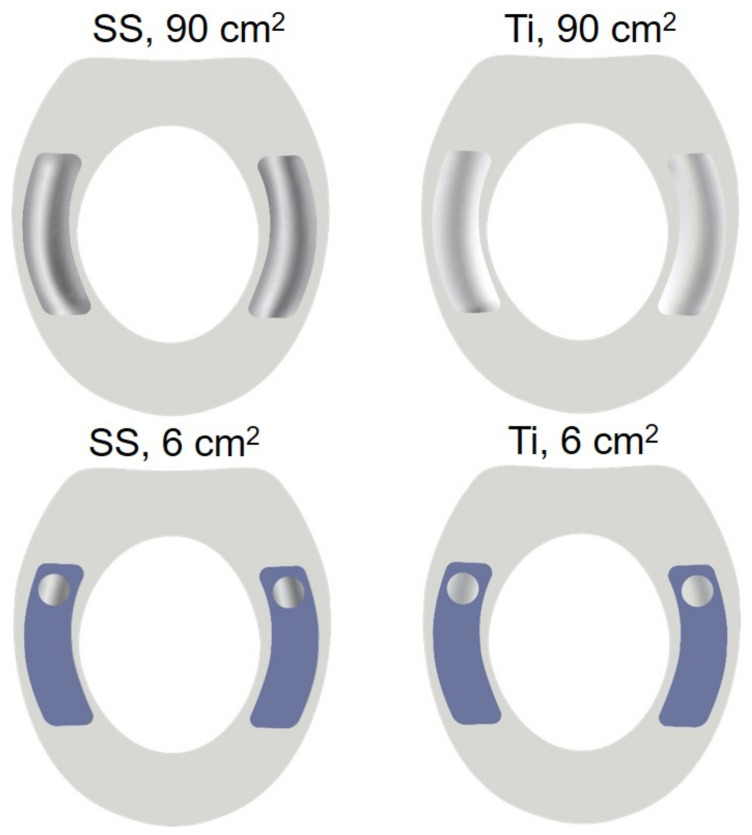
Fully integrated seat with 90 cm^2^ stainless steel (**top-left**), 90 cm^2^ titanium (**top-right**). Electrodes are covered with a vinyl mask (shown in blue), to achieve a smaller area of 6 cm^2^ of stainless steel (**bottom-left**) and titanium (**bottom-right**). For gold standard, alcohol swabs were placed on the 6 cm^2^ electrodes for both materials.

**Figure 4 sensors-22-08510-f004:**
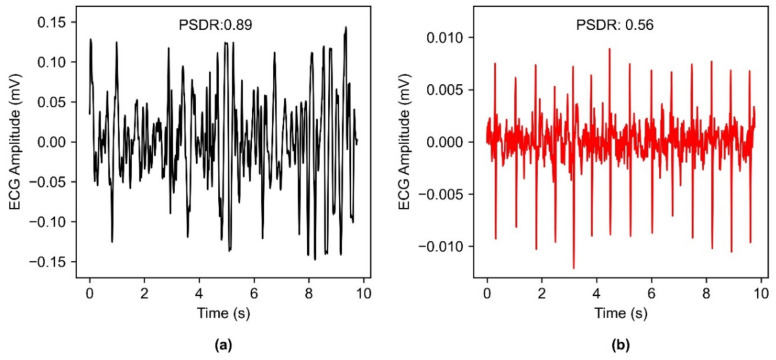
An example of the power spectral density ratio (PSDR) calculation computed for ECG signal (**a**) PSDR for a poor-quality signal with large noisy spikes results in a value of 0.89 and is characterized as unanalyzable. (**b**) Clean ECG signal results in a value of 0.56 and is considered an analyzable signal.

**Figure 5 sensors-22-08510-f005:**
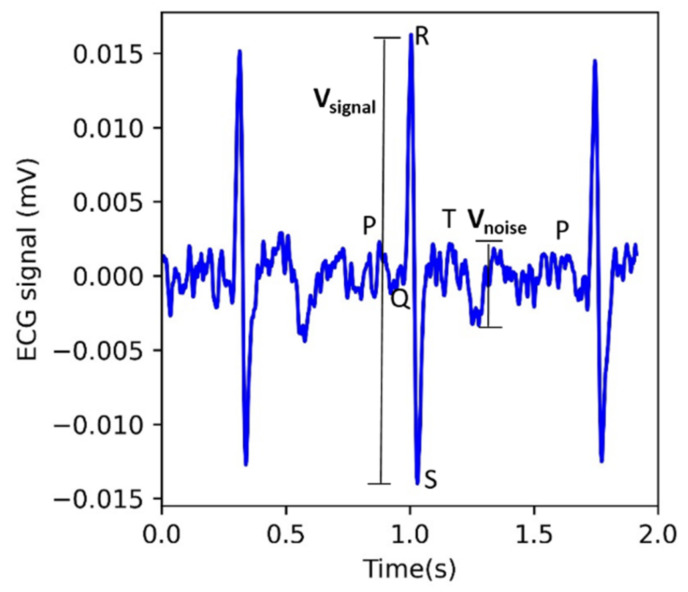
An example of signal-to-noise ratio (SNR) calculation, where *V_signal_* represents the peak-peak signal (R and S wave), and *V_noise_* represents the peak-peak noise in the segment after the T wave ends and before the P wave of the next ECG cycle begins.

**Figure 6 sensors-22-08510-f006:**
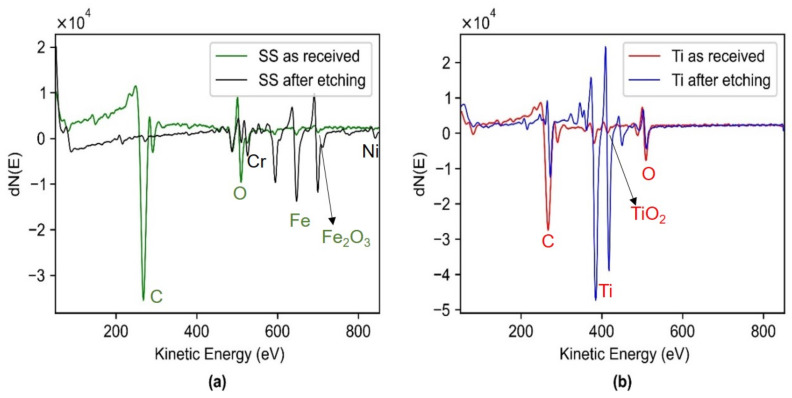
(**a**) AES profile for SS as received, which shows the presence of iron oxide (Fe_2_O_3_) along with carbon and oxygen. AES profile for SS after etching shows chromium and nickel become visible, and iron concentration increases. (**b**) AES profile for Ti as received, which shows the presence of titanium along with carbon and oxygen and the peak at 420 eV indicates the presence of TiO_2_. AES profile of Ti after etching shows the concentration of titanium increases and oxygen decreases.

**Figure 7 sensors-22-08510-f007:**
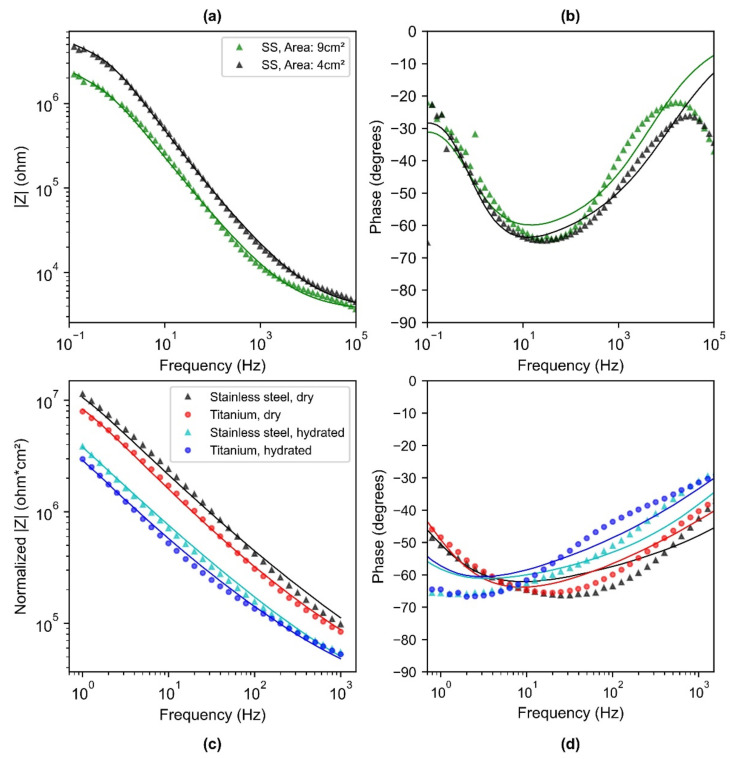
(**a**) Bode plot depicting the impedance magnitude (**b**) and phase response for SS electrode of area 9 cm^2^ and 4 cm^2^ on a dry phantom. (**c**) Bode plot depicting the normalized impedance magnitude and (**d**) phase response for SS (9 cm^2^) and, Ti (9 cm^2^) on a dry and hydrated phantom. Normalized impedance response is shown across the low-frequency range 1 Hz–1000 Hz (crucial for biopotential signals) to clearly show the differences between material and hydration in the lower frequency regime. Points represent the measured data and the solid lines represent the fitted curve obtained via the model.

**Figure 8 sensors-22-08510-f008:**
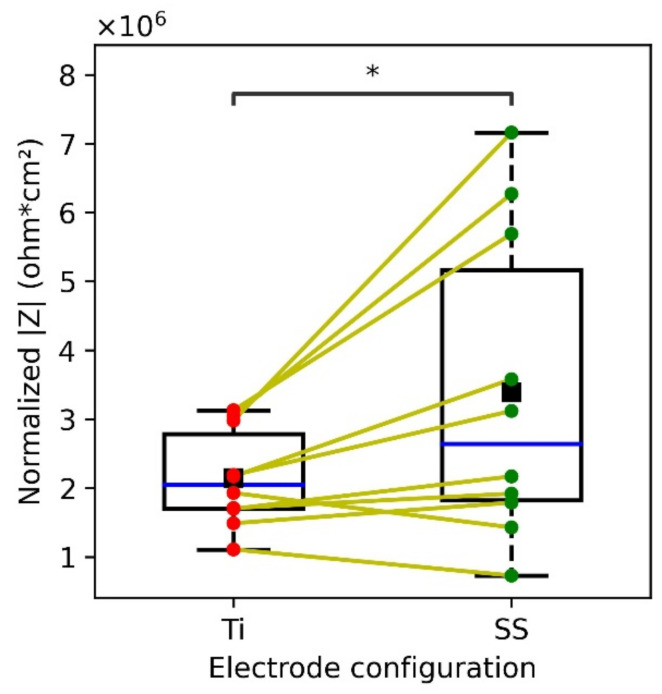
Box plot depicting the normalized skin-electrode contact impedance with respect to area, obtained with Ti and SS electrodes across ten human subjects. Data points corresponding to each subject across Ti (red dots) and SS (green dots) are connected with lines (yellow). Blue line shows the median and black square depicts the mean. The normalized skin-electrode contact impedance shown here is experimentally obtained from the impedance spectrum at 10 Hz, which depicts the combination of *R_sc_*, *C_sc_*, and *C_c_*.

**Figure 9 sensors-22-08510-f009:**
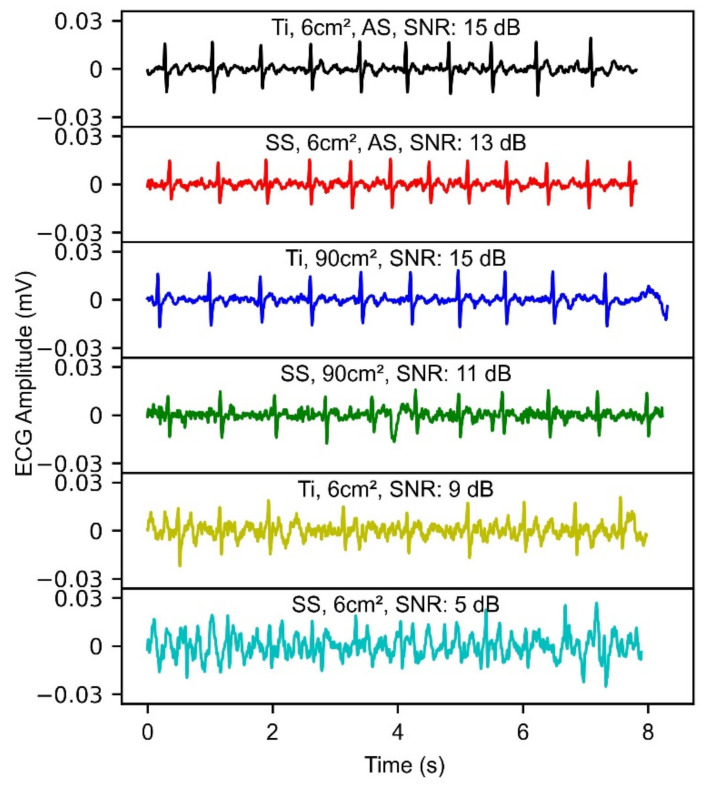
Example of the different ECG waveforms across one subject (only 8 s shown, 90-s recordings were made) using different electrode configurations, along with their computed SNR.

**Figure 10 sensors-22-08510-f010:**
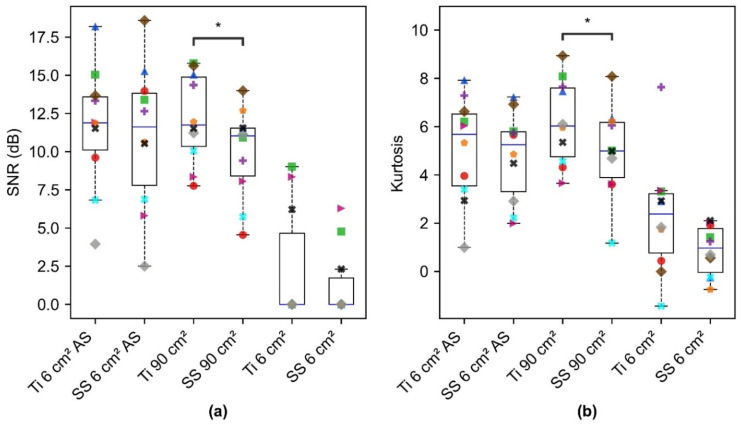
Computed ECG signal quality metric depicting the (**a**) SNR and (**b**) Kurtosis across different electrode configurations (material and area). Ti 6 cm^2^ AS and SS 6 cm^2^ AS denote the gold standard measurements, obtained with alcohol swabs placed directly on the electrodes to come into contact with the skin. Box plots show the entire distribution, data points corresponding to each subject are represented by a different shape and color. (*) denotes the statistical significance of *p* < 0.05, obtained with paired t-test and was significant for Ti 90 cm^2^ and SS 90 cm^2^. Paired t-test conducted for (Ti 6 cm^2^ AS, SS 6 cm^2^ AS); (Ti 6 cm^2^, SS 6 cm^2^); (Ti 6 cm^2^ AS, Ti 90 cm^2^); (SS 6 cm^2^ AS, SS 90 cm^2^) was not significant.

**Figure 11 sensors-22-08510-f011:**
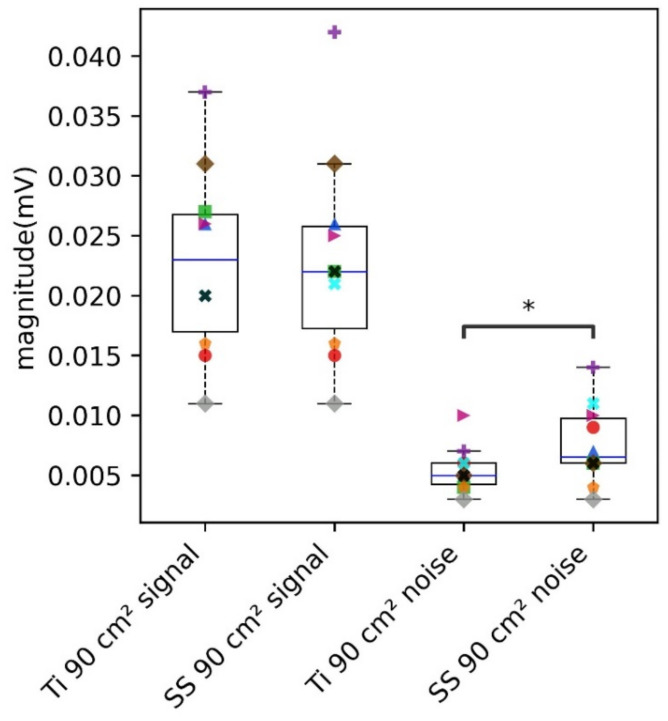
Evaluation of the contribution of signal and noise components across large areas of SS and Ti electrode configurations. Box plots show the entire distribution; the datapoint corresponding to each subject is represented by a different shape and color. (*) denotes the statistical significance of *p* < 0.05, obtained with paired *t*-test.

**Figure 12 sensors-22-08510-f012:**
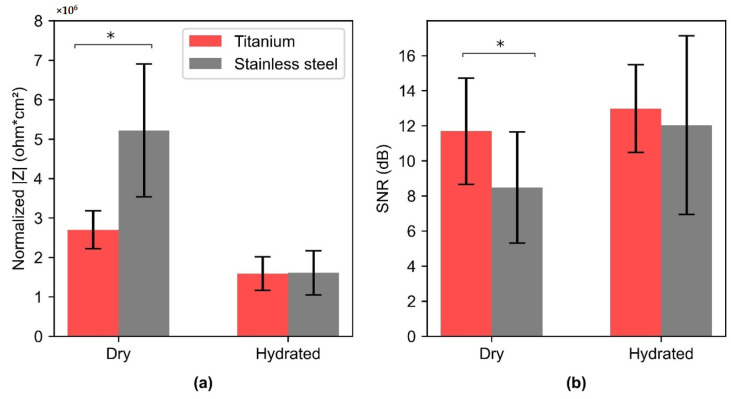
(**a**) Normalized Skin-Electrode contact impedance to the area at 10 Hz for dry (N = 5) and hydrated skin (N = 5) individuals with SS and Ti. (**b**) Obtained SNR for ECG across the dry and hydrated skin individuals with SS and Ti. (*) denotes the statistical significance of *p* < 0.05, obtained with paired *t*-test. Rectangular bars show the mean for the individuals that fall under dry and hydrated skin type, N = 5 for each category, and the height of the error bars denote the standard deviation.

**Table 1 sensors-22-08510-t001:** Equivalent impedance fitting parameters obtained for impedance response across SS electrodes of 4 cm^2^ and 9 cm^2^ area electrodes on a dry phantom. *R_d_* and *R_sc_* represent the resistance of deeper tissue layer and stratum corneum, respectively; *C_sc_* and *C_c_* represent the real capacitances of stratum corneum and contact, obtained from the fitted constant phase elements.

Electrode Area	*R_d_* (kΩ)	*R_sc_* (MΩ)	*C_sc_* (nF)	*C_c_* (µF)	Goodness of Fit
4 cm^2^	1.77	1.62	151	0.68	0.017
9 cm^2^	1.75	0.70	391	1.34	0.044

**Table 2 sensors-22-08510-t002:** Equivalent impedance fitting parameters obtained for impedance response across SS and Ti on a dry and hydrated phantom. Impedance normalized to the area is shown. *R_d_* and *R_sc_* represent the resistance of deeper tissue layer and stratum corneum, respectively; *C_sc_* and *C_c_* represent the real capacitances of stratum corneum and contact, obtained from the fitted constant phase elements.

Electrode Material (Phantom Hydration Status)	*R_d_* (kΩ cm^2^)	*R_sc_* (MΩ cm^2^)	*C_sc_* (nF cm^−2^)	*C_c_* (nF cm^−2^)	Goodness of Fit
Ti (dry)	9.3	6.4	41	267	0.007
SS (dry)	9.3	6.4	41	114	0.031
Ti (hydrated)	9.3	3.0	190	420	0.042
SS (hydrated)	9.3	3.0	190	224	0.035

## Data Availability

Not applicable.

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
