# Peer review of "Dependence of Skin-Electrode Contact Impedance on Material and Skin Hydration"

_sensors, 2022, doi:10.3390/s22218510_

Round 1

Reviewer 1 Report

This is an interesting paper studying the influence of skin hydration on the performance of dry electrodes. Two metallic dry electrodes were tested, namely titanium and stainless steel dry electrodes.

The authors should address the following topics before the paper is ready for publication:

- The authors should inform about the pressure that was used for the EIS experiments with the phantom.

- As the authors state, the CPE is very useful to take into account several characteristics of the interface, such as roughness and inhomogeneity. The authors should provide the values of the CPE parameters and draw their conclusions on the information that was possible to obtain from those values.

- In Figure 5 Vnoise should be represented by a vertical bar, just like Vsignal.

- Regarding the values of Csc and Cc disclosed in Tables 1 and 2, are they real capacitances or CPE values ? If they are CPE values, the units are different.

- In Figs 7, 10 and 12 the (a) and (b) plotted in the figures should be represented by the same letters as in the legends (not A and B)

- The goodness of the fit doesn’t validate a model by itself. The authors should also represent the phase associated with the EIS data. Furthermore, they should provide the errors associated with the calculated fitted parameters. For example, if the error is larger than the parameter itself, this means it cannot be used, and eventually the model is not correct.

Reviewer 2 Report

The authors investigated the dependence of skin-electrode contact impedance on material and skin hydration. This work provides significant insights for the development of new gel-free electrodes for biopotential acquisition. Although being interesting, I find that there are some major issues with the paper that require addressing prior to this being considered for publication in this journal. I have identified the main points for consideration below:

1.     This manuscript has minor spelling typos, style errors and grammatical errors. Please carefully check the whole manuscript.

2.     10 Hz electrode-skin impedance is an important parameter for biopotential acquisition. The area-normalized electrode-skin impedances at 10 Hz for SS and Ti electrodes are recommended to be added in the main text.

3.     The authors use a 25 mV signal do measure the impedance. This can be really excessive and may not ensure the linearity between potential and current. Signals that high are very rarely used in electrochemistry and only for very resistive systems. Typical excitation signals are of the order of 5 mV. The authors should check the linearity of the potential vs current around such high potential values by measuring the impedance with 5, 10 mV, 15 mV and 25 mV for example.

4.     How about the feasibility on recording EEG signals with SS and Ti dry electrode?

5.     A recent reference (Sensors and Actuators B 241 (2017) 1244–1255) involving the effect of the contact area on the electrode-skin impedance is recommended to be cited.

6.     The full names of acronyms like SS should be given when first mentioned.

7.     In the discussion section, I suggest that the author combine these findings on electrode-skin impedance with the design of new reliable biopotential electrodes. I’d like to hear the authors' suggestions for establishing low and stable electrode- skin impedance.

8.     Semi-dry electrode that replace conductive gels with a tiny amount of electrolyte liquid has become a typical electrode for bipotential monitoring. The tiny amount of electrolyte liquid can maintain relatively low and stable electrode-skin impedance, which is consistent with the hydration effect on the contact effect. So, I’d like to hear the discussions about semi-dry electrode. In addition, some related references  about semi-dry electrodes are recommended to be cited, such as J. Neural Eng. 18 (2021) 046016; J. Neural Eng. 17 (2020) 051004; Research, 2022, 2022, 9830457.

Round 2

Reviewer 1 Report

Data form Table S1 and S2 present CPE b values of the order of 0.5, which is an indication of a Warburg element, not a capacitor. But it is strange from to imagine a Warburg element to model the electrode/skin interface.

On the other hand, I noticed that the authors used the same electrodes as working and reference, which is not valid in this case as Ti or SS cannot be used as potential references.

My suggestion is that the authors perform the impedance measurements by using a large area, gelled Ag/AgCl electrode, as the reference electrode, which should improve the quality of the results.

Also, I think the authors shouldn’t have used the region between the QRS components of the ECG for the calculation of the noise as there are other components of the ECG signal there (see Fig.5). 

Reviewer 2 Report

The authors have well addressed all my comments. Thanks for an interesting paper.

Author Response

Thank you for your insights and comments in the first round of revisions. These helped make our paper more clear.